# Large-Size Subunit Catalases Are Chimeric Proteins: A H_2_O_2_ Selecting Domain with Catalase Activity Fused to a Hsp31-Derived Domain Conferring Protein Stability and Chaperone Activity

**DOI:** 10.3390/antiox11050979

**Published:** 2022-05-17

**Authors:** Wilhelm Hansberg, Teresa Nava-Ramírez, Pablo Rangel-Silva, Adelaida Díaz-Vilchis, Aydé Mendoza-Oliva

**Affiliations:** 1Departamento de Biología Celular y del Desarrollo, Instituto de Fisiología Celular, Universidad Nacional Autónoma de México, UNAM, Mexico City 04510, Mexico; tnava@ifc.unam.mx (T.N.-R.); prangel@ifc.unam.mx (P.R.-S.); adelaidadv@gmail.com (A.D.-V.); ayde.mendoza@utsouthwestern.edu (A.M.-O.); 2Center for Alzheimer’s and Neurodegenerative Diseases, University of Texas Southwestern Medical Center, Dallas, TX 75235, USA

**Keywords:** large-size subunit catalase, C-terminal domain, molecular chaperone, catalytic activation, heat stability, amino acid frequency, evolution of catalases, Hsp31

## Abstract

Bacterial and fungal large-size subunit catalases (LSCs) are like small-size subunit catalases (SSCs) but have an additional C-terminal domain (CT). The catalytic domain is conserved at both primary sequence and structural levels and its amino acid composition is optimized to select H_2_O_2_ over water. The CT is structurally conserved, has an amino acid composition similar to very stable proteins, confers high stability to LSCs, and has independent molecular chaperone activity. While heat and denaturing agents increased *Neurospora crassa* catalase-1 (CAT-1) activity, a CAT-1 version lacking the CT (C63) was no longer activated by these agents. The addition of catalase-3 (CAT-3) CT to the CAT-1 or CAT-3 catalase domains prevented their heat denaturation in vitro. Protein structural alignments indicated CT similarity with members of the DJ-1/PfpI superfamily and the CT dimers present in LSCs constitute a new type of symmetric dimer within this superfamily. However, only the bacterial Hsp31 proteins show sequence similarity to the bacterial and fungal catalase mobile coil (MC) and are phylogenetically related to MC_CT sequences. LSCs might have originated by fusion of SSC and Hsp31 encoding genes during early bacterial diversification, conferring at the same time great stability and molecular chaperone activity to the novel catalases.

## 1. Introduction

Monofunctional heme-catalases dismutate H_2_O_2_ into one molecule of O_2_ and two molecules of H_2_O. They are monophyletic enzymes that group into three clades: small-size subunit catalases (SSCs) (Clade 1 and 3) and large-size subunit catalases (LSCs) (Clade 2) [1,2,3]. LSCs are similar to SSCs but have an additional CT of 150–190 amino acid residues.

*N. crassa* has two LSCs, CAT-1 and CAT-3, which are the main catalase activities during the asexual life cycle [4]. CAT-1, although glycosylated [5], does not have a signal peptide and is a cytosolic enzyme [4]; by contrast, CAT-3 has four *N*-glycosylation sites [6], has a signal peptide for secretion [4], is mainly associated with but not bound to the cell wall [7], and can diffuse from the cell wall to the surrounding medium [8]. CAT-1 accumulates in cells during the stationary growth phase [4] and even more in asexual spores (conidia) where it constitutes 0.6% of total protein [5]. The extracellular CAT-3 becomes oxidized during the first 10 min of conidia germination, probably by singlet oxygen that is produced during this process [4,9]. The accumulated CAT-1 in conidia is the main catalase activity during germination and the start of exponential growth but thereafter it becomes diluted by growth; instead, CAT-3 is increasingly expressed and becomes the main activity after several hours of exponential growth [4]. CAT-3 is also induced under stress conditions, such as exogenous H_2_O_2_, Paraquat, cadmium, uric acid, nitrate or by heat shock [4].

We have found that asexual development is triggered by oxidative stress, characterized by increased production of reactive oxygen species, redox imbalance, protein oxidation and degradation, and rapid changes in enzyme activities [10,11,12]. As a response to oxidative stress, CAT-3 is induced during hyphal adhesion and development of aerial hyphae, and CAT-1 is highly expressed when conidia are formed [4]. A CAT-3 null mutant, D*cat-3*, results in increased level of asexual and sexual development [8,12].

Here, we analyzed the amino acid frequency of the catalase domain and the CT to conclude that the catalytic domain was optimized to select H_2_O_2_ over water molecules and at the same time preserved its stability by incorporating “order promoting” amino acid residues; the CT was selected to confer chaperone activity and great stability to LSCs. We investigated the stability of the catalase and the CT domains and assayed the chaperone activity of the CT on the heat resistance of the catalytic domain. Furthermore, the possible origin of the CT was analyzed by primary sequence and structural analysis with members of the DJ-1/PfpI superfamily. We found that the bacterial Hsp31 is the only protein of this superfamily that has sequence similarity with the MC of catalases. A phylogenetic analysis indicated that LSCs might have originated by a fusion of SSC and Hsp31 encoding genes during early bacterial diversification.

## 2. Materials and Methods

### 2.1. Purification of CAT-1

*N. crassa* was grown in Vogel-saccharose medium in agar slants and conidia were harvested in acetone. CAT-1 was extracted as described [5]. Briefly: Two grams of dried acetone powder were ground with glass pearls and cell extract was frozen and thawed twice. After centrifugation the supernatant was precipitated with (HPLC-quality) acetone overnight, centrifuged and the pellet was dried and resuspended in Na/K phosphate buffer 50 mM, pH 7.8, prepared by mixing 7.098 g Na_2_HPO_4_ and 0.986 g KH_2_PO_4_/L (PB). Ammonium sulfate was slowly added to the solution to 18% (*w*/*v*), centrifuged and the supernatant was brought to 36% ammonium sulfate. After centrifugation, the pellet was dissolved in 0.5 M ammonium sulfate in PB, loaded onto a Phenyl-Sepharose CL-4B column and washed with 2 column volumes of 0.5 M ammonium sulfate in PB. The protein was eluted with two volumes of 10 mM Na/K phosphate buffer, pH 7.8, and the activity peak was concentrated in an Amicon Ultrafree-CL Cellulose (30,000 NMWL Millipore) and washed three times with PB. Specific activity was determined, and the CAT-1 was stored in aliquots at −18 °C. 

### 2.2. Optimal Temperature

Catalase activity was measured at 22 °C by determining the initial rate of O_2_ production with a Clark electrode in a sealed two-walled 2 mL chamber with circulating water to control temperature [5]. Reaction was started by adding 0.04 µg of CAT-1 in 1 µL to a 2 mL stirring solution of 10 mM of H_2_O_2_ in PB. H_2_O_2_ concentration was determined by absorbance at 240 nm in a 1 cm path cell, considering an extinction coefficient of 35.76 M^−1^ cm^−1^. One unit is defined as µmoles of H_2_O_2_ consumed per min, at 22 °C, under these conditions. Data were analyzed with the OriginPro software.

### 2.3. Treatment of CAT-1 with Subtilisin

A 500 µL mixture containing 200 µg of CAT-1 and 0.2 U/mL of subtilisin in PB was incubated 30 min at 30 °C. Reaction was stopped by adding 200 mM PMSF in DMSO and cooling in ice. After centrifugation for 5 s, the supernatant was analyzed by 10% PAGE under denaturing conditions and 8% PAGE under non-denaturing conditions for in-gel catalase activity detection (Appendix A). The CT of the tetramer was completely digested by the protease giving C63, composed of 63 kDa peptides; the MW corresponds to the CAT-1 monomer without the CT. The peptide concentration was determined at 280 nm with an extinction coefficient of 88,810 M^−1^ cm^−1^. After PAGE with 8% acrylamide under non-denaturing conditions, the C63 band was cut into small pieces and the enzyme was eluted from the gel in 5 mL PB for 1 h. After centrifugation, the C63 was concentrated in a 30 kDa Amicon filter.

### 2.4. Saturation Kinetics of the CAT-1 Catalytic Domain

Catalase activity was measured at 22 °C by determining the initial rate of O_2_ production with a Clark electrode. Reaction was started by injecting catalase into a two-walled chamber with circulating water to control temperature. The sealed chamber was filled with 2 mL of a known concentration of H_2_O_2_ in PB. Data were fit to a two-components Hill equation [13]. Non-enzymatic H_2_O_2_ decomposition (base line) was subtracted from each determination.

### 2.5. Determination of t_1/2_

CAT-1 and C63 were incubated in 3 mL of PB in a thermo-block at the desired temperature. Aliquots of 2 mL (0.5 µg) were transferred to a two-walled chamber with a Clark electrode and incubated for 1 min at 22 °C before reaction was started by adding 10 mM H_2_O_2_ and run for 3 min. Initial velocity was determined.

### 2.6. Treatment of CAT-1 with Denaturation Agents

Equilibrium was determined with each detergent used, which was reached after 48 h. No further change was detected after 72 h incubation. CAT-1 was incubated for 72 h in 2.5 mL with different concentrations of either SDS, CTAB or TX100 in PB at 25 °C. Thereafter catalase activity was determined at 25 °C by oximetry using 2 mL solution starting the reaction by adding 10 mM H_2_O_2_.

CAT-1 and C63 were incubated with increasing concentration of guanidinium ion at 25 °C and activity was measured thereafter by oximetry. CAT-1 was incubated for 12, 24 and 72 h in 2.5 mL with different concentrations of guanidinium ion in PB at 25 °C. Then, 2 mL solution was taken to measure catalase activity at 25 °C. 

### 2.7. Expression of Proteins in Escherichia coli

CAT-3, CAT-3 without its CT (CAT-3^DTD^), and the CT of CAT-3 (TDC3) were cloned in pCold^TM^ I, expressed in *E. coli* and purified by a 6×His tag affinity column as described [14].

### 2.8. Structural Alignment

The CT structure of CAT-1 and CAT-3 was used to perform a structural alignment with all proteins in the PDB using the DALI server v.3 [15]. Proteins with the highest structural similarity were two other catalases and members of the DJ-1/PfpI superfamily. Fifteen proteins having different reported functions were selected from this superfamily. A multiple alignment with the 19 proteins was carried out using the POSA server v.1 [16]. Thereafter a distance matrix of the apparent root-mean-square deviation (RMSD) and a phenogram using the D-UPGMA server were elaborated [17]. The RMSD is an apparent value because of different resolution of each protein structure (between 1.05 and 2.3 Å).

### 2.9. Heat Stability and Molecular Chaperone Activity

Heat stability was determined from 40 to 90 °C for 1 h and followed by light scattering at 360 nm in a Beckman DU650 spectrophotometer. CAT-1, CAT-3, C63 and CAT-3^ΔTD^ were used at 3 µM, calculated as tetramers; TDC3 was added at 3 µM and 6 µM, calculated as dimer.

### 2.10. Dimerization Modes of CT

Structure alignments were performed using the PDBeFold program. Residues that interact in the interface of the two CT monomers of CAT-3 were determined using the online tool PDBePISA [18]. PDB access number: DJ-1: 1ucf, YhbO: 1io4, Hsp31: 1izy, YDR533Cp: 1qvw, CAT-3: 3ej6, CAT-1: 1sy7, *Penicillium vitale* (*P. janthinellum*) catalase (PVC): 2xf2, HPII: 5bv2, and *Mycothermus*
*thermophilus* catalase: 5zz1.

### 2.11. Phylogeny of Bacterial Hsp31 and MC_CT

About 220 bacterial HchA sequences were downloaded from public databases and aligned with Clustal Omega multiple sequence alignment. Sequences that were overrepresented in some bacterial families (e.g., Enterobacteriaceae, Pseudomonadaceae) and those that were very similar to another were eliminated. The 186 selected sequences were aligned with three groups of bacterial and two of fungal MC_CTs (Appendix A). To verify its relatedness, HchA sequences that clustered with all the MC_CTs were aligned with each individual group of MC_CTs. From the cluster of HchA sequences, published information about the isolation or habitat of each bacterial strain was retrieved.

## 3. Results

### 3.1. Amino Acid Composition of the Catalase Domains and CTs

The analysis of catalase domains from 38 different SSCs and LSCs without their CT (Appendix A) shows a peculiar composition of amino acid residues when compared with the SwissProt Database statistics mean frequency of amino acids: hydrophobic amino acids are reduced (15–19%), hydrophilic amino acids (T, S, Q, N, R, K, D, E) are similar in total percentage (43–44%) but some residues (D, N, T, R) are augmented while others (E, Q, S, K) are reduced, and heterocyclic/aromatic amino acids (H, P, F, W, but not Y) are greatly augmented (30–50%), in both the heme pocket and the whole catalase domain (Table 1 and Appendix A).

Cys and Met are hydrophobic amino acids that generally have a decreased frequency in catalases (C, −64%; M, −22%); many catalases lack Cys altogether. However, the occurrence of other hydrophobic residues is greatly reduced in catalases such as Leu (−30%) and Ile (−29%) (Appendix A).

Some hydrophilic amino acid residues predominate over others: Asp (+28%) is preferred over Glu (−16.5%), Asn (+20%) over Gln (+10%), Thr (+16%) over Ser (−23%), and Arg (+4%) over Lys (−16%) (Appendix A). In the catalase domains, there is a clear skewed ratio of Asp/Glu (from the mean of 0.81 in PDB to 1.21), Asn/Gln (from 1.03 to 1.19), Thr/Ser (from 0.81 to 1.05), and Arg/Lys (from 0.95 to 1.21) (Table 1).

The incidence of heterocyclic/aromatic amino acid residues is greatly augmented in the catalytic domain of catalases: there is a great increase in Phe (+84%), Trp (+82%), His (+74%) and Pro (+42%), but not significantly in Tyr (+6.8%) (Appendix A). 

Besides a similar low contend of Cys and Met, the CT has a completely different frequency of amino acid residues: total hydrophobic residues are augmented (13–29%), especially Ala, Val and Gly; total hydrophilic amino acids are diminished (10–25%), especially Asn, Gln and Glu, and heterocyclic/aromatic residues are diminished (12–22%), particularly Pro (−27%), whereas Phe (+28%) is augmented (Table 1 and Appendix A). The CT is very stable according to its amino acid composition. Stability was confirmed with the IUPred3 prediction program [19] (Appendix A).

### 3.2. The CT Confers Overall Protein Stability and Chaperone Activity

The CT confers stability to HPII of *E. coli*, which is heat stable up to 95 °C, and heat stability is lost when the CT is digested away with a protease [22]. The protease processed HPII turns the protein instable and favors its aggregation [23]. Additionally, a truncated version of *Staphylococcus aureus katA* lacking the CT has a reduced catalase activity because of instability, causing a decrease in growth of the bacterium under oxidative stress [24].

Besides giving stability to the catalase domain, the CT has a molecular chaperone activity that prevents denaturation of other proteins, by heat, urea, and H_2_O_2_ [14]. The TDC3 also protects SSCs from heat denaturation [14]. The structure of the CT is conserved in all LSCs and is similar to Hsp31 and DJ-1 proteins, which are well known chaperones. CAT-3 or TDC3 increases survival of *E. coli* under heat or oxidative stress conditions, whereas the CAT-3 without the CT domain does not [14]. The last 17 amino acid residues, a mobile coil that is rich in charged and hydrophobic residues, determines molecular chaperone activity of CAT-3: absence of this coil abolished, and mutant variants of the coil diminished chaperone activity [25].

CAT-1 is very resistant to heat and presence of denaturing agents [5]. We assayed the catalase activity after incubating the enzyme at different temperatures or treating it with different detergents or guanidinium ion, which are known denaturing agents. There is a linear increase of specific activity (0.14 × 10^−6^ U/mg/°C) with temperature from 5 to 55 °C, and thereafter an abrupt decrease for the next ten degrees followed by a slower reduction in activity up to 80 °C (Figure 1).

CAT-1 was treated with subtilisin to degrade the CT. After treatment, the active tetramer was composed by 63 kDa peptides, indicating digestion of the whole CT (Appendix A). The subtilisin digested CAT-1 tetramer (C63) preserved its catalase activity (Figure 2). The initial rate, obtained in the range of 10 mM to 3 M H_2_O_2_ concentration, was fit to the equation that results from adding two Hill equations [13]. Similar to the untreated enzyme, saturation kinetics of C63 did not follow Michaelis–Menten but presented two-components [5,13]. The first component gave a S_0.5(1)_ (K_M_) of 0.039 M, a V_max(1)_ of 2.9 × 10^6^ U/mg, and a Hill number of 1.2; the second component gave a S_0.5(2)_ of 0.489 M, a V_max(2)_ of 3.6 × 10^6^ U/mg, and a Hill coefficient of 1.7. Values are similar to the untreated enzyme [13], indicating that the digestion of the CT with subtilisin did not affect the catalytic domain of the enzyme.

When the purified native CAT-1 was incubated at 93 °C for different times, after 10 min the enzyme activity decreased slowly and almost linearly within 1.5 h, giving a t_1/2_ of 72 min; by contrast, the C63 activity decreased exponentially with a t_1/2_ of 16 min (Figure 3). This result indicates that the CT confers CAT-1 its high thermal stability.

When the CAT-1 was treated with different concentration of detergents at equilibrium, the activity increased 60% with 1% SDS, 2% with TX-100, and 120% with 2% C-TAB, whereas no increase in activity was obtained with the C63 (Figure 4).

When CAT-1 was incubated in the presence of guanidinium ion for 24, 48 and 72 h, there was a 5.4-fold increase in activity with 3 M guanidinium ion in 24 h, diminishing with further incubation time or higher concentration of guanidinium ion. The C63 at 24 h incubation increased its activity 3-fold with 2 M, but activity was lost with further incubation time at all concentrations (Figure 5).

CAT-1, CAT-3, and CAT-1 and CAT-3 without CT (C63, CAT-3^DTD^) were assay for their heat stability determined by light scattering. When temperature was increased in steps of 5 °C, C63, CAT-3^DTD^ behaved similarly: light scattering increased linearly between 50 and 90 °C in comparison with CAT-1 and CAT-3 that start to denature only at 90 °C (Figure 6). This indicates that the catalase domain is stable up to 50 °C and that the high thermal stability of LSCs is conferred by the CT.

To assay if the TCD3, having molecular chaperone activity, increased the stability of the catalase domain, TCD3 was added in two concentrations, in a 1:1 TDC3:C63 (or CAT-3^DTD^) ratio (as in the normal tetramer) or in a 1:2 ratio. As seen in Figure 6, TDC3 can protect both catalase domains from heat denaturation in a concentration dependent manner.

### 3.3. Structural Alignment of the CT with the Members of the DJ-1/PfpI Superfamily

The sequence similarity of the CT of LSCs is low (22.4–26.9%) but the structure is conserved and is similar to Hsp31 and DJ-1 [14]. Fifteen structures of representative members of the DJ-1/PfpI superfamily were downloaded from the PDB and a structural alignment with the CT of the four known LSCs structures was carried out. As seen in Figure 7, all members of the DJ-1/PfpI superfamily aligned to the CTs with and an apparent RMSD of 1.4–3.3 Å, although the catalase orthologues PVC and CAT-3 were more similar to Hsp31 from yeast and *Candida albicans*, and HPII (*katE*) and CAT-1 more like DJ-1 and YhbO-type proteins.

### 3.4. CT of LSCs Represents a New Dimer Conformation of the DJ-1/PfpI Superfamily

Members of the DJ-1/PfpI superfamily have been classified according to how the monomers interact to form a dimer. Four dimerization types are described: DJ-1 (I), YhbO (II), Hsp (III) and YDR (IV) [26]. In each case, the interaction (I–IV) involves the same structures in both monomers, giving a symmetrical conformation. Region I (DJ-1) comprise: α1, β4, the N-terminal of α7, the C-terminal of α8 and two loop regions, which are between β3 and β4, and between β11 and α7 (Figure 8A). Region II (YhbO) involves: α2, α4 and α5 (Figure 8B). Region III (Hsp): 21 residues of the N-terminal, the region from β4 to β5 and the loops between β6 and α3 (Figure 8C). *E. coli* Hsp31 has an extra loop of 45 residues between regions I and II that also interact in the dimer. Region IV (YDR): β7, and the region from β8 to β9 (Figure 8D).

LSCs have two dimers of CT, one in each pole of the tetrameric structure. Regions that interact in the CT dimer are different to the four regions described. In LSCs the region, now called patch V, includes α22 helix of CAT-3 (which correspond to α3 and α4 in the DJ-1 structure) the loop between α22 and α23 and the C-terminal loop (Figure 8E). Other LSCs present the same interacting regions (α22 in CAT-3 corresponds to α26 in CAT-1, α30 in PVC, α32 in HPII and α28 in *M. thermophilus*) and in all of them the C-terminal loop is involved, although it adopts two different conformations (Figure 8F).

### 3.5. Possible Origin of the CT from a Bacterial Hsp31-like Protein

*E. coli* has three GSH independent glyoxalases which might also be protein deglycases [27]: HchA corresponds to the Hsp31 protein, YhbO is similar in sequence to two proteases: Intracellular protease and Protease I, and YajL is orthologous to DJ-1/Park7 proteins. These proteins together with transcription factor AraC conform the DJ-1/PfpI superfamily. The CT of LSCs is also part of this superfamily and is reported as a branch of the AraC sequences [28]. In fact, when an alignment is carried out with several sequences of each of these proteins, the AraC group comes out as a sister group of the CT of LSCs. However, this is not due to a general resemblance of the CT of LSCs to AraC. HchA, YhbO, Intracellular protease, Protease I, YajL together with the CT of bacterial and fungal LSCs have three analogous contiguous hydrophobic sequences flanked or intermingled with charged residues, whereas AraC does not have these three contiguous sequences but aligns with different parts of the protein (Appendix A).

Alignment of AraC sequences with a file consisting of the MC plus the CT (MC_CT) of bacterial and fungal LSCs (Appendix A) indicated that only 48.4% of the file was recognized as similar. AraC had the lowest similarity with the file of 43 MC_CT sequences. Alignment of the MC_CTs file, which includes two different bacterial groups plus two *Streptomyces* groups– which have an extra sequence–, and two fungal groups (L1 and L2) with DJ-1 and YajL sequences gave a 65.4% similarity; with YhbO and Intracellular protease, 60%, and with Protease I, 71%. With HchA (Hsp31) sequences, 82.4% of the MC_CT sequences were recognized as related. These differences are because the MCs were recognized as analogous to the HchA proteins, but not the other proteins.

The MC joins the helical domain to the CT domain in LSCs. We included this sequence (31 amino acid residues) because we recognized that it was similar to the N-terminal domain of the bacterial Hsp31 proteins; the other DJ-1/PfpI proteins did not show likeness with the MC of LSCs (Appendix A); fungal Hsp31 do not have this N-terminal region. 

A phylogeny was realized with 186 HchA sequences from different bacterial phyla and families (10 Bacillota, 2 Deinococcus Thermus, 2 Cyanobacteria, 33 Actinomycetota, 6 Bacteroidota and Pseudomonadota: 2 delta, 26 alpha, 28 beta, 76 gamma). In total, 57 sequences preferentially aligned with the fungal L1 group, recognizing a sequence that is not present in other MC_CTs, 17 aligned preferentially with L1 and the group of bacterial MC_CTs that is closest to L1, and 14 with the fungal L2 group. A group of five bacterial HchA sequences that clustered with all MC_CTs includes: one Bacillota (Firmicutes) (*Exiguobacterium aestuarii*), one Actinomycetota (Acidimicrobiia) two Bacteroidota and one Pseudomonadota delta (Appendix A). Another group of 22 HchA sequences, that is always associated with the set of MC_CT sequences, consisted of five Actinomycetota, two Bacteriodota, Pseudomonadota: one delta, three alpha, 10 gamma. Three other sequences were also related: two Bacteroidota and one Campylobacterota (Appendix A). Because the related bacterial Hsp31 sequences stem from many bacterial phyla the probable fusion of SSC and Hsp31 encoding genes probably occurred very early in the bacterial phylogeny before the diversification of extant phyla.

During the search for HchA sequences, glutamine amidotransferase domain-containing protein (GATD) sequences were retrieved. GATD proteins are closely related to Hsp31 but not to other members of the DJ-1/PfpI superfamily; however, they do not align with the MC of LSCs.

## 4. Discussion

Stability of proteins greatly depends on the content of hydrophobic amino acid residues [29]. The hydrophobic residues Cys and Met are prone to oxidation by H_2_O_2_ [30,31] which could be the reason why catalases and catalase-peroxidases [32] have low amounts of these amino acid residues. However, the occurrence of other hydrophobic residues, is also reduced in the catalase domain, such as Leu (−30%) and Ile (−29%). In general, a 15–19% lower frequency of hydrophobic residues was detected in 38 SSCs and the LSCs without the CT, belonging to different groups of bacterial and fungal catalases (Appendix A). Due to the complex entrances and channel system [33,34], catalases are similar to sponges that soak water containing H_2_O_2_ to guide its substrate through the conic entrances to its deep-buried active site [3]. Being H_2_O_2_ a hydrophilic substrate, most areas of these entrances are hydrophilic [3]. Hydrophobic residues could be decreased in catalases due to the large areas occupied by the hydrophilic entrances and channels.

A low number of hydrophobic residues compromises the stability of proteins [29]. Considering that catalase stability could be affected because of a low content in hydrophobic residues, one would expect the selection of “order-promoting” amino acids over “disorder-promoting” residues [21]. In fact, the skewed ratio in catalases of Asp/Glu (from the mean of 0.81 in PDB to 1.21), Asn/Gln (from 1.03 to 1.19), Thr/Ser (from 0.81 to 1.05), and Arg/Lys (from 0.95 to 1.21) can be explained by the fact that Asp, Asn, and Thr are order-promoting amino acids, while Glu, Gln, and Ser are disorder-promoting amino acids, and Lys is more disorder-promoting than Arg [21] (Table 1 and Appendix A).

The increase in Phe (+84%) and Trp (+82%) can be explained by both, for being hydrophobic and order promoting amino acids. However, Tyr is hardly increased (+6.8%) and is also hydrophobic and an order-promoting amino acid. The difference between Phe or Trp and Tyr is that the first two have an increased residence time for H_2_O_2_ compared with H_2_O while the second does not. Moreover, the catalase domain has an increased number of His (+74%) which is a low order-promoting amino acid and Pro (+42%) which is a high disorder-promoting amino acid. Other *N. crassa* enzymes that use H_2_O_2_ as a substrate have increased amounts of Trp and Pro [32]. We have put forward that the enrichment of these amino acid residues in catalases is due to the differences in residence time of H_2_O_2_ compared to water molecules in the vicinity of these amino acid residues [33]. This molecular dynamic study gave the following sequence of differences in the residence time of H_2_O_2_ compared with water molecules in the first and second solvent shell: K > R = W > C = H > M > P = E = D > F > Y. The residence time variance of H_2_O_2_ compared with water molecules is 3.5-fold higher in Trp and Arg compared to Tyr. This would imply that catalases were selected to increase the content of Trp, His, and Pro, Asp and Arg as a resource to select H_2_O_2_ molecules in a sea of water. In fact, the entrances to the gate are rich in these amino acid residues (and low in hydrophobic residues) [33]. There is no doubt that H_2_O_2_ is selected over water: in the final section of the substrate channel, water molecules leave the channel five times faster than H_2_O_2_ molecules [33,35].

Other factors that promote stability in proteins are its length [36] and probably the network of hydrogen bonds. Catalases are relatively large proteins (>480 amino acids) and therefore would be stable proteins just by size. It is however reveling that SSCs of plants require a chaperone to maintain stability and activity. In *Arabidopsis thaliana* and rice, the activity of all three SSCs depends on the specific interaction with the chaperone NCA1 (No Catalase Activity 1) [37,38]. Furthermore, a thioredoxin acts as a molecular chaperone for catalase-3 (AtCAT3) and other proteins in the peroxisome matrix of *A. thaliana* [39]. Furthermore, the CT of LSCs is a molecular chaperone for other proteins including SSCs. When the TDC3 is expressed in an *E. coli* strain that lacks catalase, it confers a higher survival capacity to the bacterium under heat and H_2_O_2_ stress [14].

In contrast to the catalase domain, the CT has a completely different frequency of amino acid residues. Total hydrophobic residues are augmented, especially Ala, Val and Gly, which are high in very stable proteins [29]. Total hydrophilic amino acids are diminished (10–25%), especially Asn, Gln and Glu, the last two are also diminished in very stable proteins [29]. Heterocyclic/aromatic residues are decreased (12–22%), particularly Pro (−27%), which is a high disorder-promoting residue, whereas Phe (+28%) is augmented, increasing the stability of this domain.

The CT confers LSCs their great stability. Having a CT dimer in each pole of the tetramer gives rigidity to the whole structure. In fact, the b-values of CAT-1 and CAT-3 are very low, except for the N-terminal end, the MC, some small loops in the C-terminal domain, and the C-terminal ending loop in CAT-3 [6,31,38].

Heat and presence of detergents or guanidinium ion are conditions which loosen the structure of CAT-1. Increasing the mobility of amino acid residues would favor the diffusion of the substrate to the active site allowing an increase in enzyme activity. Optimal activity in CAT-1 is 55 °C, which is high compared to most fungal enzymes, indicating that heat progressively loosens the structure up to a point where the active site starts to be affected or water molecules increasingly compete with H_2_O_2_ for the active site. C63, in which the CT was degraded with subtilisin, is less stable and heat rapidly inactivated the enzyme; detergents did not affect the activity, and the guanidium ion partially activated the enzyme at a lower concentration but was inactivated with incubation time in all concentrations. All these data demonstrate that the CT gives rigidity and stability to CAT-1.

Because the CT of CAT-3, TDC3, has a molecular chaperone activity, we assayed its effect on the catalase domains of CAT-1 and CAT-3 at different temperatures. TDC3 at a 1:1 ratio completely restored the stability and at a lower ratio (1:2) the stability was only partially increased. These results indicate that TDC3 protects the catalytic domain from heat denaturation in a concentration dependent manner.

A structural alignment showed that proteins from the DJ-1/PfpI superfamily have a remarkable structural similarity with the CT of LSCs according to the low apparent RMSD (1.4–3.3 Å). PVC and CAT-3 (orthologues) CT structure is more similar to the fungal (RMSD 2.3 Å) than the bacterial Hsp31 proteins (RMSD 3.3 Å), probably because the bacterial proteins are larger (285 amino acids) than the fungal (240 amino acids). The CT structure of HPII (*katE*) and CAT-1 is more similar to YhbO-type proteins (RMSD 1.9 Å) (172 amino acids) and DJ-1 proteins (RMSD 2.5 Å) (190 amino acids).

The DJ-1/PfpI superfamily dimeric proteins have been grouped into four modes of interaction between monomers (patch I-IV). The symmetric dimer of the CT domains at the poles of the tetrameric structure of the LSCs constitute a new mode of interaction (patch V). It is notable that different modes of interaction to form symmetric dimers have molecular chaperone activity, such as DJ-1, Hsp, and the CT of LSCs, suggesting that a symmetric dimer might be related to chaperone activity.

When the sequences of the different proteins of the DJ-1/PfpI superfamily were compared with the CTs, the result clearly indicated sequence similarity between all of them. HchA, YhbO, Intracellular protease, Protease I, YajL and the CT have three contiguous hydrophobic sequences flanked or intermingled with charged residues which indicate sequence relatedness. In contrast AraC does not have these sequences but has many stretches of two to four hydrophobic amino acid residues flanked by charged residues which gives such a sequence many possible ways to align with the other members of the DJ-1/PfpI superfamily. From all these proteins, the bacterial Hsp31 showed higher similarity (82%) with all MC_CTs because bacterial Hsp31 proteins have a N-terminal sequence that is similar to the MC of LSCs. Some bacterial Hsp31 sequences also recognize a specific region of the CT of Streptomyces and the fungal L2 LSCs. 

To assure that the sequence relationship between the MC_CT of different LSCs with the bacterial HSP31 was common, a phylogeny was generated with 186 HchA sequences from different bacterial phyla and families. All these sequences aligned with the MC_CT of the different groups of LSCs, 57 sequences aligned preferentially with the fungal L1 group and another 17 with L1 and its closest bacterial group. These 74 HchA sequences recognized a specific sequence which is present only in L1 group MC_CTs. 

The MC_CT of each group of LSCs probably evolved differently losing parts of the original bacterial Hsp31-like protein: bacterial Hsp31 is 285 amino acid residues long, the MC_CT has 192 amino acid residues in Streptomyces, 182 in other bacteria, 213 in L1 and 180–190 in L2 fungal LSCs. Although each CT evolved differently, some regions conserved similarity to the bacterial Hsp31 proteins and tend to be present in all four groups, even though sequence similarity between bacterial and fungal is only 53–56%.

Because the 30 HchA sequences that consistently associated with all MC_CTs stem from bacteria belonging to different phyla, we concluded that the possible fusion between SSC and Hsp31 encoding genes occurred very early on the bacterial phylogeny, before the diversification of extant phyla. This is consistent with the fact that LSCs are present in all these phyla and that Clade 1 SSCs appeared before LSCs during evolution [40]. It is notable that the HchA sequences related to all MC_CTs include the 6 Bacteroidota and the 2 Pseudomonadota delta sequences but none of the 28 beta. Additionally, what is notorious is the marine extreme habitat of most of these bacteria (Appendix A) which could also suggest an ancient origin.

## 5. Conclusions

LSCs probably originated by the fusion of a bacterial SSC gene and a Hsp31-type gene, which conferred two great advantages for bacterial survival: a great stability to the catalase domain which became resistant to different stress conditions, albeit with a lower catalytic activity, and a molecular chaperone activity, for other proteins including SSCs.

## Figures and Tables

**Figure 1 antioxidants-11-00979-f001:**
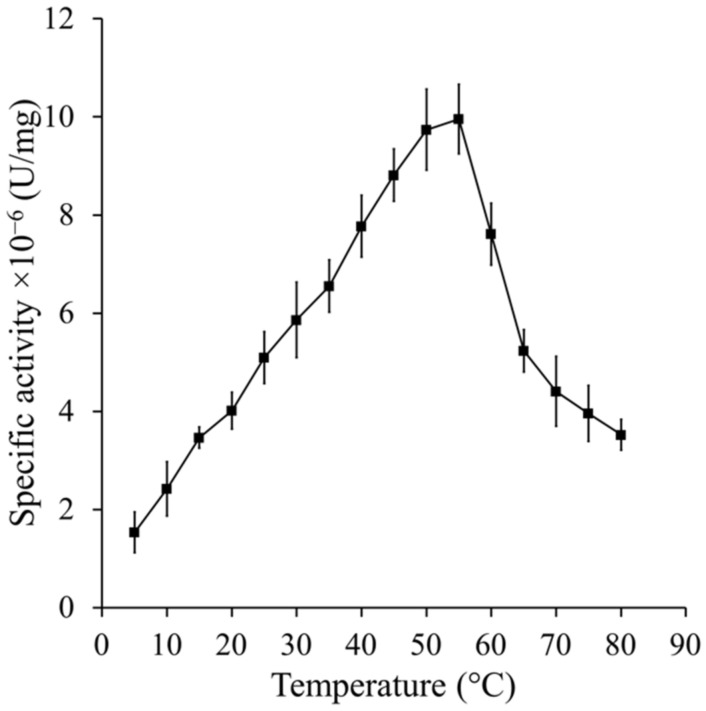
Optimal temperature of CAT-1. Specific activity was determined by oximetry at temperatures from 5 to 80 °C in increasing steps of 5 °C degrees in a sealed double-walled chamber with circulating water at the adjusted temperature. Reaction was started by adding 0.04 µg of CAT-1 to a 2 mL solution of 10 mM H_2_O_2_ in PB. Units are defined as µmoles of H_2_O_2_ consumed/min under these conditions.

**Figure 2 antioxidants-11-00979-f002:**
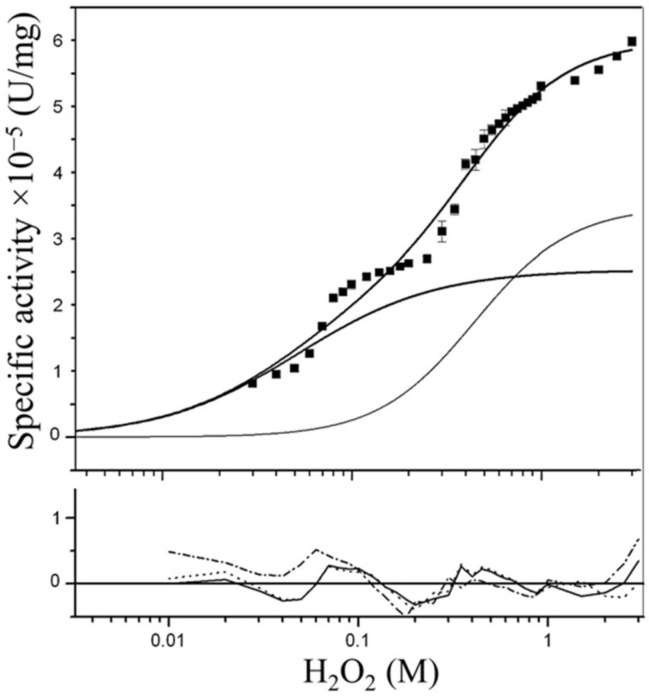
Saturation kinetics of the CAT-1 catalytic domain (C63). Data are the mean ± SEM of three separate determinations. Solid lines: fits of the data to the two-components Hill equation with Hill numbers of 1.2 and 1.7 and theoretical curves for each component. Shown below the graph are the residuals from the fit of the data to the three equations considered: solid lines, two-component equation with Hill numbers of 1.2 and 1.7 for the first and second component, respectively; dot lines, two-component equation without cooperativity; dash dot lines, single Michaelis–Menten equation.

**Figure 3 antioxidants-11-00979-f003:**
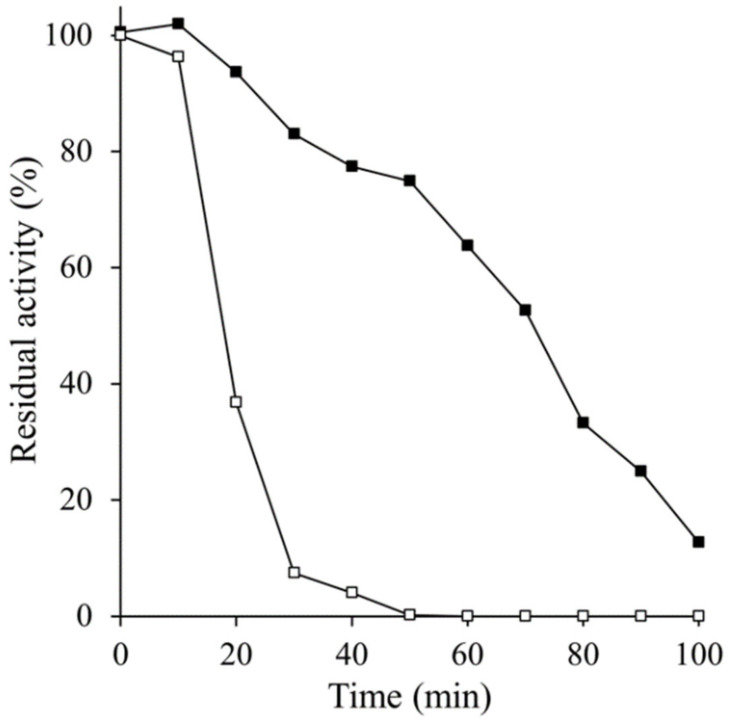
Residual activity during heat denaturation. CAT-1 (closed squares) or C63 (open squares) was incubated at 93 °C, and residual activity was measured by oximetry every 10 min.

**Figure 4 antioxidants-11-00979-f004:**
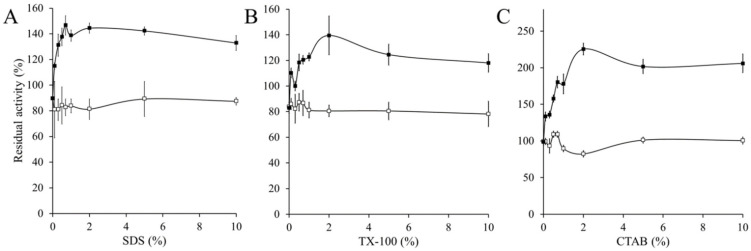
Increase in catalase activity in the presence of detergents. CAT-1 (closed squares) and C63 (open squares) were incubated for 72 h, at 25 °C, in the presence of different concentrations of either (**A**) SDS, (**B**) TX-100 or (**C**) CTAB in PB, pH 7.8. Thereafter catalase activity was determined by oximetry at 25 °C, starting the reaction by adding 10 mM H_2_O_2_.

**Figure 5 antioxidants-11-00979-f005:**
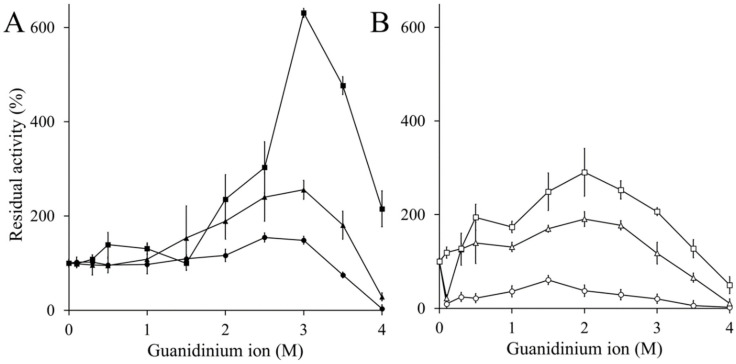
Catalase activation by guanidinium ion. (**A**) CAT-1 and (**B**) C63 enzymes were incubated with increasing concentration of guanidinium ion for 24 (closed and open squares), 48 (closed and open triangles) and 78 h (closed and open rhomboids). Thereafter catalase activity was measured by oximetry.

**Figure 6 antioxidants-11-00979-f006:**
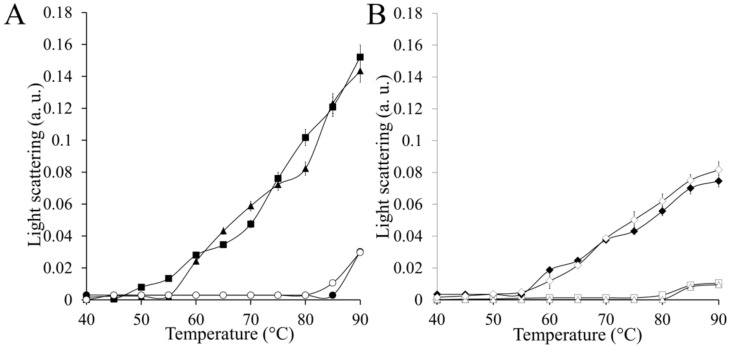
(**A**). Heat stability of CAT-1 and CAT-3 and their catalase domains. Labels: 3 µM CAT-1 (opened circles), 3 µM CAT-3 (closed circles), 3 µM C63 (closed triangles), and 3 µM CAT-3^∆TD^ (closed squares). (**B**). Heat stability of the catalase domains in the presence of the CT chaperone TDC3. Labels: 3 µM C63 plus 6 µM TDC3 (opened triangles), 3 µM CAT-3^∆TD^ plus 6 µM TDC3 (opened squares), 3 µM CAT-3^∆TD^ plus 3 µM TDC3 (opened rhomboids), and 3 µM C63 plus 3 µM TDC3 (closed rhomboids). CAT-3, the catalase domain CAT-3^∆TD^, and TDC3 were expressed in *E. coli* and purified [14]. Proteins were incubated at different temperatures for 60 min and light scattering was followed at 360 nm (a. u., arbitrary units). Each point is the mean of three different experiments.

**Figure 7 antioxidants-11-00979-f007:**
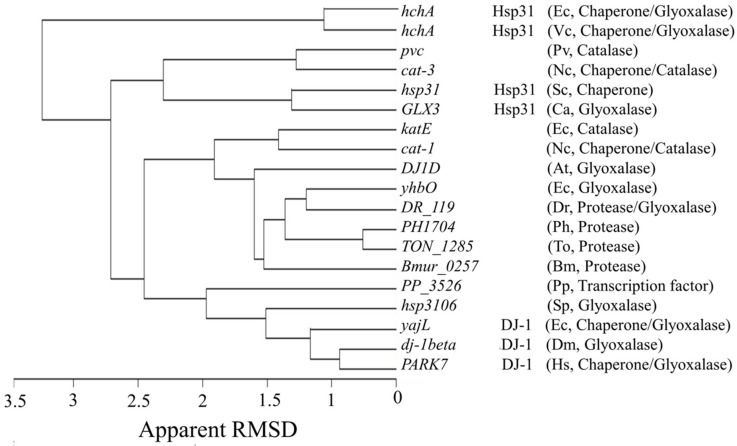
Structural alignment of four CTs with representative structures of the DJ-1/PfpI superfamily. Structures of 15 different members of the DJ-1/PfpI superfamily having various functions were aligned with the CT of 4 LSCs. Structure resolution of these proteins varied between 1.05–2.3 Å. Gene and proteins names, in the case of DJ-1, Hsp31 are indicated, and, in parenthesis, the organism and function(s) reported. Ec, *E. coli*; Vc, *Vibrio cholerae*; Sc, *Saccharomyces cerevisiae*; Ca, *Candida albicans*; Nc, *N. crassa*; Pv, *P. vitale* (*P. janthinellum*); Pp, *Pseudomonas putida*; Dr, *Deinococcus radiodurans*; Ph, *Pyrococcus horikoshii*; To, *Thermococcus onnurineus*; At, *Arabidopsis thaliana*, Bm, *Brachyspira murdochii*; Sp, *Schizosaccharomyces pombe*; Dm, *Drosophila melanogaster* and Hs, *Homo sapiens*.

**Figure 8 antioxidants-11-00979-f008:**
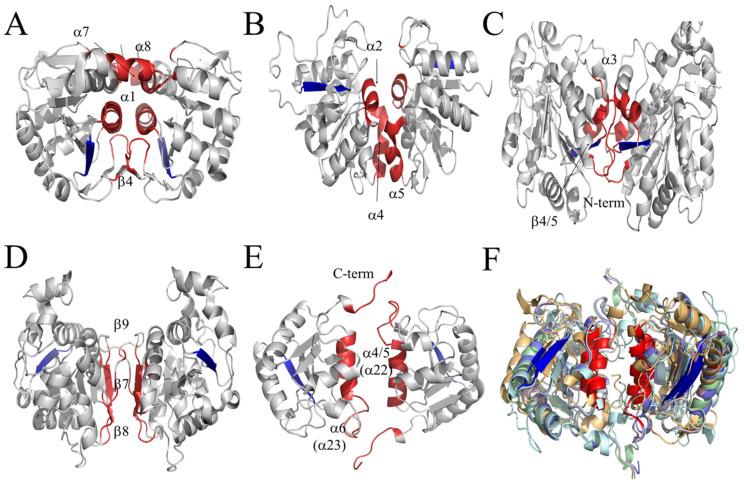
Dimerization modes of members of the DJ-1/PfpI superfamily. (**A**) The DJ-1-type dimer, patch I. (**B**) The YhbO-type dimer, patch II. (**C**) The Hsp-type dimer, patch III. (**D**) The YDR-type dimer, patch IV. (**E**) The CT dimer of LSCs represent a new type, patch V. (**F**) Superposed CT dimers of LSCs: CAT-1 (pale cyan) and CAT-3 (pale blue) of *N. crassa*; PVC (pale green) of *P. janthinellum* (*P. vitale*), HPII (pale orange) of *E. coli,* and CATPO (pale pink) of *M. thermophilus*. The interacting regions (patches) involved in the interaction are marked in red. β1 is marked in blue to figure out the orientation of the monomers.

**Table 1 antioxidants-11-00979-t001:** Amino acids frequency (%) of the catalase domain (CAT) and the CT domains when compared with the mean value of the PDB. CATs have decreased hydrophobic, altered ratio of hydrophilic (D/E, N/Q, T/S, R/K) and increased heterocyclic/aromatic amino acid residues [(F + W)/P]. CTs have increased amounts of hydrophobic residues and lower numbers of hydrophilic and heterocyclic/aromatic amino acids. hpho, hydrophobic, hphi, hydrophilic, and hcy/aro, heterocyclic/aromatic amino acid residues. Hydrophobicity scale for amino acids was taken from measurements of these residues at the protein surface [20]. IF = Instability Factor, % of each amino acid multiplied by the Disorder Propensity value of each residue [21]. ^a^ Higher value means decreased disorder propensity. ^b^ Higher value indicates increased disorder propensity. ^c^ Mean values of the CTs of fungal LSCs (L1- and L2-type, 25 orthologues each) (Appendix A).

	PDB	CAT	CT ^c^
hpho ^a^	41.53	35.28	49.14
hphi	43.49	43.82	37.22
hcy/aro	14.88	20.73	13.35
D/E ^a^	0.81	1.21	0.80
N/Q ^a^	1.03	1.16	1.07
T/S ^a^	0.81	1.05	0.78
R/K ^a^	0.95	1.22	0.83
(F + W)/P ^a^	1.05	1.16	1.56
IF ^b^	3.87	4.11	−0.04

## Data Availability

The sequences of catalases and Hsp31 are available from the NCBI GenBank k (https://www.ncbi.nlm.nih.gov/genbank/) and (https://www.uniprot.org/) accessed on 10 April 2022.

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
