# Peer review of "Large-Size Subunit Catalases Are Chimeric Proteins: A H2O2 Selecting Domain with Catalase Activity Fused to a Hsp31-Derived Domain Conferring Protein Stability and Chaperone Activity"

_antioxidants, 2022, doi:10.3390/antiox11050979_

Round 1

Reviewer 1 Report

The study presented in this manuscript appears to be based on a solid background, and its conclusion are well argued. I have only found some minor points requiring Authors' intervention:

Line 74 “dyed” should be changed into “dried” (I suppose…)

Line 143 “o” should be changed into “to”

L 183 PRO is NOT an aromatic amino acid

L 188 PHE is NOT a heterocyclic amino acid

L 281 Please check this ambiguous sentence

L 457 Please check grammar

Author Response

Errors have been corrected and sentences clarified.

To include Pro and Phe in a group, the expression heterocyclic/aromatic amino acids is now used: Lines: 377, 389, 395, 557, 558, 559, 1020, Table 1 and in Table S2

Reviewer 2 Report

In this paper, Hansberg et al show the role of the C-terminal domain (CT) present in bacterial and fungal large-size catalases, regarding both the catalase activity and stability. Also, CT seems to be related to the DJ-1/PfpI superfamily, suggesting that the bacterial protein Hsp31 might be the origin of the chaperone function of CT in evolution. The research is sound and well conducted, so it can be published in Antioxidants with some revision of minor points.

Firstly, English should be revised by -preferably- a native speaker. Several errors are found throughout the manuscript. Secondly, the authors should assign another name to “Tm”, since traditionally, in biophysics, Tm is referred to melting temperature, and not the half-life time of enzyme activity. And thirdly, the authors (or editors) should be aware that in page 10 of the manuscript the Greek symbols as alpha or beta might be missing (lines 333 to 336); if not, that paragraph does not make sense.

Author Response

English was revised and many changes have been made.

Tm has been substituted by t1/2: Lines: 304, 653, 652